# YAMU: Yet Another Modified U-Net Architecture for Semantic Segmentation

**Pranab Samanta**                                    PRANAB.SAMANTA@AIRAMATRIX.COM

**Nitin Singhal***                                    NITIN.SINGHAL@AIRAMATRIX.COM
*Advanced Technology Group, AIRAMATRIX PVT. LTD., Mumbai, India*

## Abstract

Digital histopathology images must be examined accurately and quickly as part of a pathologist's clinical procedure. For histopathology image segmentation, different variants of U-Net and fully convolutional networks (FCN) are state-of-the-art. HistNet or histopathology network for semantic labelling in histopathology images, for example, is one of them. We improve our previously proposed model HistNet in this paper by introducing new skip pathways to the decoder stage to aggregate multiscale features and incorporate a feature pyramid to keep the contextual information. In addition, to boost performance, we employ a deep supervision training technique. We show that not only does the proposed design outperform the baseline, but it also outperforms state-of-the-art segmentation architectures with much fewer parameters.

**Keywords:** Digital histopathology, deep learning, semantic segmentation, HistNet, MRcSE

## 1. Introduction

The gold standard method of diagnosis, especially in cancer, is a pathologist's examination of patient tissue samples under a microscope. The manual technique, on the other hand, is time-consuming, labor-intensive, and necessitates a quantitative evaluation of pathology slides that can be subjective (Gurcan et al., 2009).

Computational Pathology was born with the development of digital Whole Slide Image (WSI) scanners, which use machine learning techniques to assist pathologists in their diagnosis (Burlutskiy et al., 2019; Deng et al., 2020). Specifically, WSI pixels are divided into malignant and benign zones, allowing a diagnosis to be made on the tissue sample. Deep learning models are particularly well suited for these problems since they can develop expert comprehension after being trained on annotated WSIs. Typically, pathologists scan slides for regions-of-interest (ROI) that are relevant to the disease being diagnosed, and then check these regions for abnormal tissues. Computational pathology and deep learning algorithms have shown considerable potential in highlighting (segmenting) diagnostically relevant regions as a visual aid and clinical decision support for pathologists (Webster and Dunstan, 2014; Litjens et al., 2016; Campanella et al., 2018).

*Related Works:* Medical image analysis has relied on pixel-based (Minaee et al., 2021) and entity graph-based (Ahmedt-Aristizabal et al., 2021) techniques for semantic segmentation. However, only pixel-based analysis for proper delineation of object boundaries is

---

* Corresponding author

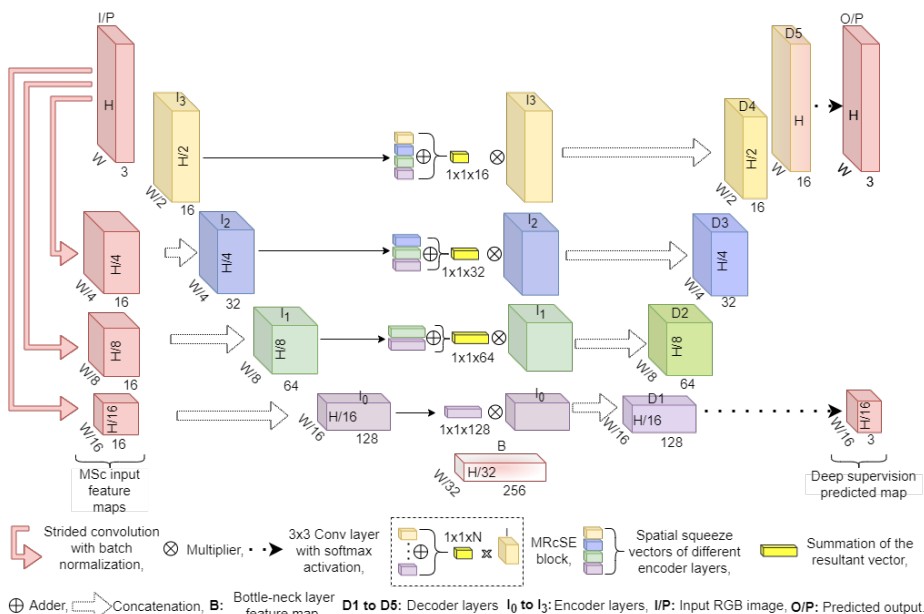

Figure 1: YAMU: Annotations and detailed structure. Multiscale input feature maps (in the leftmost column) are generated from the input feature map (I/P) in the proposed network. The feature map from the strided convolution of the stem layer is $l_3$ encoder layer. Three multi-resolution spatial squeeze and excitation blocks are included in the long skip-pathways (MRcSE).

included in the work. Contextual links and global data are used by both PSPNet (Zhao et al., 2017) and DeepLabV3+ (Chen et al., 2018). PSPNet employs a pyramid pooling module. However, because it is only used at the end of the encoder, access to multi-scale information in separate encoder-decoder stages is not possible. HistNet (Samanta et al., 2021) provides multi-scale information via the encoder stage, allowing for continuous access to multi-scale data. Works in (Zhou et al., 2018; Bilinski and Prisacariu, 2018; Zhang et al., 2018; Huang et al., 2017) use dense shortcut connections and dense skip pathways in the decoder, allowing each decoder block to access data generated by all preceding blocks. This enables the collection of background data and the combination of features from different resolutions. HistNet also allows multi-resolution feature fusion, but instead of appending output from all previous decoders, the models can choose which feature to propagate farther up the decoder at each stage.

In this work, we improve our previously proposed model HistNet (Samanta et al., 2021) by adding new skip pathways to the decoder stage to aggregate multiscale data and incorporating a feature pyramid to keep the contextual information. To improve performance even further, we use a deep supervision training technique. The proposed architecture is shown in Figure 1. We show that the proposed design outperforms not just the baseline, but also state-of-the-art segmentation architecture with far less learnable parameters.

## 2. METHOD

### 2.1. Model Architecture

Figure 1 shows the detailed architecture of our proposed model, named YAMU. By including trainable multi-scale input feature maps (MSc) in each of the three encoder layers, we improve on our prior recommended design HistNet (Samanta et al., 2021). Also, to calibrate the encoder layers features, a unique long skip pathways called multi-resolution spatial squeeze and excitation (MRcSE) are devised from encoder to decoder by mixing high resolution local features with low resolution global features. The proposed segmentation model is made up of three parts: a stem, an encoder, and a decoder, each of which contains inception+ blocks and modified dense blocks. The methodology also employs a deep supervision training strategy.

#### 2.1.1. Encoder and Decoder

The encoder block in the proposed YAMU-Net is made up of a stem block and three inception+ blocks. The stem block employs convolution (kernel size of $7 \times 7$) with stride 2 and a maxpooling layer to downscale the input image by 4, as shown in Figure 5 (a). The inception+ block (Figure 5 (b)) follows the stem block and is made up of $1 \times 1$, $3 \times 3$, and $5 \times 5$ parallel convolution layers with a short skip connection. Furthermore, the convolution technique is the same for the $3 \times 3$ and $5 \times 5$ branches. To extract information from two distinct FoVs, Inception+ blocks use a parallel of standard and dilated convolution. The model uses three alternative dilation rates of 2, 3, and 4 for the three inception+ blocks in the dilated convolution. As seen in HistNet (Samanta et al., 2021), the dilation rates are optimally found.

YAMU-net uses a customised decoder block, Figure 5 (c) to finely identify morphological aspects in a WSI. It's made up of five modified dense blocks and employs dilated convolution (dilation rate of 2). A short skip link exists in each redesigned dense decoder block, allowing features from the same and lower resolution stages to be reused. In addition, two sets of parallel and conventional convolution are used to integrate the multi-level features.

By lowering the number of channels in each of the HistNet's convolution layers, we were able to reduce the total number of learnable parameters. We divide the number of output channels by 2, 4, and 8 to get the best pruned model. The number of output channels in HistNet were 32, 32, 64, 128, 256 at the encoder layer, and 128, 64, 32, 32, 32 at the decoder layer.

Proposed supervised semantic segmentation model uses a stem block in the initial encoder layer. Some spatial information may be lost at the deeper encoder layers as a result of employing the stem block to make the model more computationally efficient. This may affect the segmentation performance. In this work, we implemented multi-scale input images at the encoder layers to address this issue.

#### 2.1.2. multi-scale input feature maps (MSc):

To keep the contextual information in the proposed model, we concatenate the local information from the input image to the deeper encoder layer, in addition to the multiscale feature extractor block, as illustrated in Figure 1. Kernel sizes of $7 \times 7$, $15 \times 15$, $15 \times 15$, and corresponding stride factors of 4, 8, and 16 with zero padding of 3, 6, and 6 are used to

compute multi-scale input feature maps of size $128 \times 128$, $64 \times 64$, and $32 \times 32$, respectively. To scale the input image, we used a convolution-based technique rather than traditional pooling operations. (Springenberg et al., 2014) presents a convolution-based technique replaces inter-feature dependencies, making the scaling operation trainable rather than fixed by backpropagation. As described in the following subsection, the study also built a new long skip connection that adaptively emphasises the essential channels utilising higher and lower resolution feature maps, namely MRcSE, and propagates it to the decoder layer.

### 2.1.3. MULTI-RESOLUTION SPATIAL SQUEEZE AND EXCITATION (MRCSE)

The new long skip connection adaptively avoids less relevant channels, as shown in Figure 2, by reconstructing the encoder layer feature based on high resolution global and low resolution local features. $1 \times 1$ convolution is employed to recalibrate the feature matrix, followed by a spatial squeeze block. Before running the resulting vector via the Spatial Squeeze (SS) block, the convolution layer is utilised to minimise the number of channels. There are four layers in the encoder (see Figure 1). The encoder layers are denoted as $\mathbf{E} = [l_0, l_1, l_2, l_3]$, with $l_3$ being the stem layer and $l_2$, $l_1$, $l_0$ being the inception+ blocks. The output channels of the indicated layers are 128, 64, 32, and 16, respectively, from $l_3$ to $l_0$. To compute the summation of the resultant vector for an encoder layer, we use $1 \times 1$ convolution for other layers as a bottleneck to maintain dimensionality and then compute the resultant vector for all layers as illustrated in Figure 2. The fused resultant vector of the $i^{th}$ encoder layer $l_i$ is made up of the total of resultant vectors formed by the local and high level global feature layers, namely $l_i$, $l_{i-1}$,..., $l_0$. To excite and transfer the output to the decoder layer, the $l_i$ encoder layer output is multiplied with this fused resultant vector.

*Computation of recalibrated/excited feature map*: assume the input feature map after $1 \times 1$ convolution layer $F_L = [f_{(L,1)}, f_{(L,2)},...,f_{(L,N)}]$ as a combination of channels $f_{L,i} \in \Bbbk^{H_L \times W_L}$. $N$ is the number of output channels in the $L^{th}$ encoder layer. Spatial squeeze is implemented by a global average pooling layer, producing vector $z_L \in \Bbbk^{1 \times 1 \times N}$ with its $k^{th}$ element given by

$$z_L(k) = \frac{1}{H_L \times W_L} \sum_{i}^{H_L} \sum_{j}^{W_L} f_{(L,k)}(i,j) \tag{1}$$

This operation embeds global spatial information from the $L^{th}$ encoder layer in vector $z_L$. The vector $z_L$ is transformed to $\hat{z_L} = \mathbf{W}_1(ReLU(\mathbf{W}_2 z_L))$, where $\mathbf{W}_1 \in \Bbbk^{N \times N}$, $\mathbf{W}_2 \in \Bbbk^{N \times N}$ are the weights of two fully-connected layers. This represents the channel-by-channel dependencies. By passing it via a sigmoid layer $\sigma(\hat{z_L})$, the dynamic range of $\hat{z_L}$ activations is brought to the interval $[0, 1]$. The resultant vector is

$$\hat{R} = [\sum_{l \subseteq E} \sigma(\hat{z}_L(1)), \sum_{l \subseteq E} \sigma(\hat{z}_L(2)), ..., \sum_{l \subseteq E} \sigma(\hat{z}_L(N))]. \tag{2}$$

$$\hat{F}_{cSE} = [f_{(L,1)} \sum_{l \subseteq E} \sigma(\hat{z}_L(1)), f_{(L,2)} \sum_{l \subseteq E} \sigma(\hat{z}_L(2)),$$
$$..., f_{(L,N)} \sum_{l \subseteq E} \sigma(\hat{z}_L(N))]. \tag{3}$$

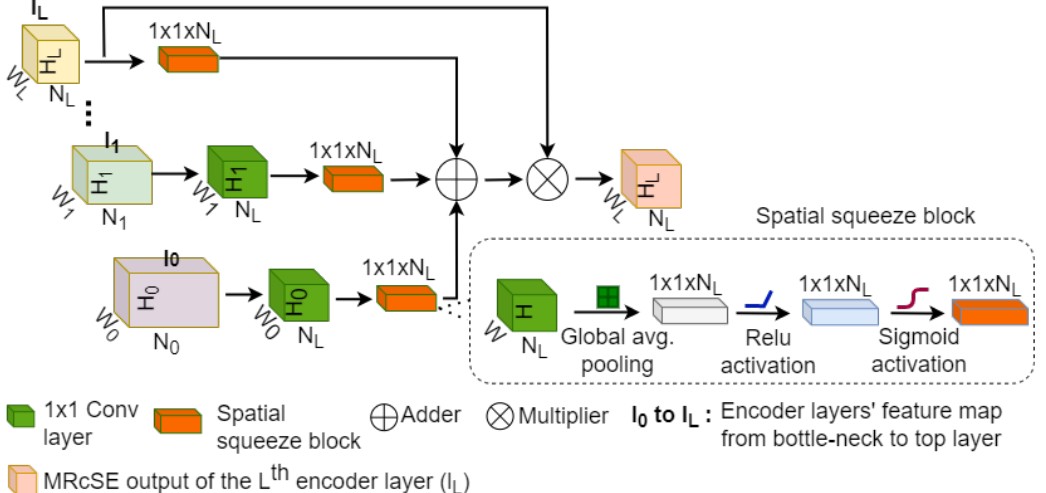

Figure 2: For the $L^{th}$ encoder layer, the long skip connection has been redesigned (MRcSE). It entails creating spatial feature maps for each channel based on its relevance. The highlighted box indicates the Spatial squeeze block used to construct the resulting vector from the input features.

This summation of the resultant vector in equation 2 is used to recalibrate the $L^{th}$ encoder layer features $F_L$ represented as $\hat{F}_{cSE}$ in equation 3. The activation $\sigma(\hat{z}_L(i))$ indicates the importance of the $i^{th}$ channel of $L^{th}$ encoder layer which are re-scaled. As the network learns, these activations are adaptively tuned to ignore less important channels and emphasize the important ones. The architecture of the block is illustrated in Figure 2.

## 2.2. Deep Supervision

In this research, we present YAMU training with deep supervision. This training method was inspired by Deeply Supervised Nets (DSN) (Lee et al., 2014; Li et al., 2019; Le'Clerc Arrastia et al., 2021). To speed up training convergence, the DSN network governs each layer based on the ground truth label. The objective is to force the output of the decoder layers to the original segmentation map. To do so, variance is added to the overall loss by combining the losses from the two decoder dense blocks' output with the appropriate downsampled version of the ground truth segmentation map. For obtaining lucidity and robustness of the information collected in the centre of the network, losses are estimated in the middle and end of the network from the $1^{st}$ and last decoder layer output, as illustrated in Figure 1. The DSN technique also aids in the solution of the vanishing gradient problem.

## 3. EXPERIMENTS

### 3.1. Datasets

This study makes use of the Prostate Cancer Grade Assessment (PANDA) dataset from the PANDA Challenge 2020 (Bulten et al., 2022; PANDA, 2020) and the Colonoscopy Tissue Segment dataset from DigestPath 2019 (DigestPath, 2019).

### 3.1.1. PANDA DATASET 2020

The PANDA dataset categorizes cancer tissue into Gleason patterns (3, 4, or 5), based on the architectural growth patterns of the tumour from histopathology scans of prostate biopsy samples (Bulten et al., 2022; PANDA, 2020). There are six class labels in the mask image. Background (non-tissue area), stroma (connective and non-epithelium tissue), benign epithelium, cancerous epithelium (Gleason 3), cancerous epithelium (Gleason 4), and cancerous epithelium (Gleason 5) are the six labels in the annotation mask. For the ablation study experimentation, we separated the data into three categories: background (stroma and non-tissue region), benign (healthy epithelium), and malignant epithelium (combination of the three Gleason patterns). And for the state-of-the-art comparison, we have used mentioned six labels annotations. Since the organizer did not publicly release the test set for blind validation. Therefore, we solely used the PANDA 2020 training dataset supplied from Radboud University Medical Center (RUMC). The PANDA 2020 dataset from the Karolinska Institute lacks pixel-level labels and was therefore excluded from training and testing. The data was divided into two groups at random: development and testing. Based on a 75 : 25 split, the development set was further divided into training and validation. To preserve balance between classes, patches/tiles were taken from the development set. We double-checked that patches from the training and validation sets came from mutually exclusive whole slide images. The study uses 13268 (from 3922 biopsies) for training and 1676 (from 600 biopsies) for testing patches with a size of 1024×1024.

### 3.1.2. Colonoscopy Tissue Segment-DigestPath2019

The DigestPath 2019 dataset is intended for binary segmentation tasks that separate benign and malignant tissue (DigestPath, 2019). The DigestPath 2019 dataset contains 660 tissue imaging samples from 324 WSI. To deal with the massive image size, we extract 12,922 tiles with a size of 512×512, of which we use 70% for training and 30% for validation. Training and validation datasets are mutually exclusive in WSI level.

## 3.2. Ablation Study

We study each stage's segmentation performance using the PANDA dataset to assess the contributions of each stage in our proposed approach. The research investigates the amount of experimentally optimal output channels, U-Net++ (Zhou et al., 2020) skip pathways vs. revised long skip pathways, multi-scale input, and deep supervision training technique. The study starts with HistNet (Samanta et al., 2021), a semantic labelling context aggregation network, as a baseline and gradually upgrades the model. We initially reduce the number of output channels in the encoder and decoder convolution layers by half to compare performance with the baseline. Table 1 compares the performance of three different pruned HistNet models to the original one. The model with 1.85 million parameters outperforms the model with 3.5 million parameters, as demonstrated.

Following that, the impact of various skip pathways is examined in Table 2. U-Net++ skip pathways provide similar performance to the original HistNet skip pathways. When we incorporated our proposed long skip pathways, the model improved 1.37 percent in the benign class, 0.85 percent in the malignant class, and overall mean improvement of 1.08 percent with lower standard deviation. We used a deep supervision training technique,

| HistNet (# parameters in millions) | Background | Benign | Malignant | Mean DICE |
|---|---|---|---|---|
| P0 (3.5) | 98.89 | 77.12 | 87.52 | 87.84 ± 8.89 |
| P1 (1.85) | 98.87 | 76.72 | 87.16 | 87.59 ± 9.04 |
| P2 (0.47) | 98.59 | 74.60 | 83.94 | 85.71 ± 9.86 |
| P3 (0.12) | 98.23 | 68.47 | 78.54 | 81.75 ± 12.54 |

Table 1: Using the PANDA2020 dataset, the effects of channel pruning on HistNet. The smaller versions of HistNet P0 are P1, P2, and P3.

| Architecture | | | | | Performance (DICE in %) | | | | # param (in M) |
|---|---|---|---|---|---|---|---|---|---|
| PC | SPW | MSc | MRcSE | DS | Background | Benign | Malignant | MD | |
| ✓ | - | - | - | - | 98.87 | 76.72 | 87.16 | 87.59 ±9.04 | 1.855 |
| ✓ | ✓ | - | - | - | 98.79 | 77.7 | 87.08 | 87.89 ±9.44 | 2.109 |
| ✓ | - | ✓ | - | - | 98.86 | 78.25 | 87.44 | 88.18±9.22 | 2.115 |
| ✓ | - | - | ✓ | - | 98.89 | 78.09 | 88.01 | 88.67 ±8.09 | 2.093 |
| ✓ | - | - | - | ✓ | 98.57 | 75.46 | 84.15 | 86.06 ±9.53 | 1.859 |
| ✓ | - | - | ✓ | ✓ | 98.89 | 79.29 | 88.18 | 88.79 ± 8.01 | 2.130 |
| ✓ | - | ✓ | ✓ | ✓ | **98.89** | **80.49** | **88.2** | **89.19 ±7.54** | **2.185** |

Table 2: An ablation study is conducted using the PANDA2020 dataset. PC stands for pruned channel model (P1), SPW for U-Net++ skip pathways, MRcSE for our proposed long skip pathways, DS for deep supervision training approach, MSc for multi-scale input feature map, and MD for mean DICE.

which resulted in a 1.28 percent increase in the benign class. The proposed redesign of the skip connection with deep supervision enhances overall performance much more.

Multi-scale features have also been added at various encoder layers, which improve model performance by 3.77 percent, 1.04 percent, and 1.16 percent for benign, malignant, and overall mean DICE, respectively.

In several pruned versions of the HistNet, we also strive to demonstrate the utility of redesigned long skip connections using deep supervision training technique. It demonstrates how the modified long skip connection may be used in a smaller network efficiently. Tables 3 shows that redesigned long skip connections and deep supervision are more successful in a smaller network.

| Model (# parameters in millions) | Background | Benign | Malignant | Mean Dice |
|---|---|---|---|---|
| P0 + MRcSE + DS (4.01) | 98.88 | 79.75 | 88.21 | 88.94±8.22 |
| P1 + MRcSE +DS (2.13) | 98.89 | 79.29 | 88.18 | 88.79 ± 8.01 |
| P2 + MRcSE + DS (0.54) | 98.59 | 75.26 | 84.60 | 86.15 ± 9.58 |
| P3 + MRcSE + DS (0.14) | 98.26 | 70.34 | 80.79 | 83.12 ± 11.52 |

Table 3: Effects of Multi-resolution spatial squeeze and excitation (long skip connection) with deep supervision (DS) using the PANDA2020 dataset on different pruned version of the HistNet.

| Model | Backbone | Class 0 | Class 1 | Class 2 | Class 3 | Class 4 | Class 5 | # params in millions |
|---|---|---|---|---|---|---|---|---|
| FCN-8s (Long et al., 2014) | - | 93.71 | 92.80 | 68.86 | 62.29 | 71.29 | 53.32 | 69.75 |
| DeepLabV3+ | Xception | 96.55 | 94.44 | 71.81 | 67.96 | 73.45 | 50.23 | 41.25 |
| PSPNet (Zhao et al., 2017) | VGG-16 | 95.01 | 93.38 | 61.18 | 53.85 | 65.52 | 38.22 | 17.11 |
| SegNet (Badrinarayanan et al., 2017) | VGG-16 | 93.84 | 93.43 | 58.42 | 47.95 | 61.02 | 24.57 | 11.55 |
| DeepLabV3+ (Chen et al., 2018) | ResNet-50 | 91.04 | 90.33 | 65.52 | 59.63 | 66.18 | 48.60 | 10.38 |
| U-Net++ (Zhou et al., 2018) | VGG-19 | 95.04 | 94.06 | 70.35 | 61.28 | 67.87 | 55.80 | 09.05 |
| U-Net (Ronneberger et al., 2015) | VGG-19 | 66.05 | 62.13 | 15.15 | 22.08 | 29.51 | 15.19 | 07.76 |
| DeepLabV3+ | EfficientNet-B0 | 94.38 | 93.07 | **78.27** | 72.80 | 79.13 | 45.29 | 06.41 |
| MSU (Abraham and Khan, 2018) | - | 94.79 | 94.03 | 69.33 | 57.59 | 67.22 | 22.76 | 05.64 |
| ResCU-Net (Shen et al., 2020) | - | 95.25 | 94.77 | 76.26 | 68.11 | 75.70 | 40.83 | 40.73 |
| HistNet (Samanta et al., 2021) | - | 95.98 | 95.07 | 74.31 | 68.38 | 74.57 | 57.57 | 03.5 |
| **YAMU** | - | 95.91 | **95.29** | 78.14 | **73.72** | **79.50** | **64.25** | **02.18** |

Table 4: Accuracies of segmentation models on Radboud prostate tissue segment dataset-PANDA2020. Class:0, 1, 2, 3, 4, 5 corresponds to the background, stroma, benign, Gleason grade 3, Gleason grade 4, and Gleason grade 5, respectively.

| Model | Mean DICE (%) | Model | Mean DICE (%) |
|---|---|---|---|
| FCN-8s | 91.85 | U-Net++ | 85.69 |
| DeepLabV3+ | 90.13 | MSU | 89.12 |
| PSPNet | 90.54 | ResCU-Net | 92.12 |
| SegNet | 90.43 | HistNet | 92.36 |
| U-Net | 85.43 | **YAMU** | **93.50** |

Table 5: Results on colonoscopy tissue segment dataset- DigestPath2019.

### 3.3. Discussion

The results of many popular DL-based segmentation models on the Radboud PANDA dataset for Gleason grading in prostate core needle biopsies are summarised in Table 4. On this data, our proposed model has a relative increase of around 12.15% over the initial FCN-8s model. In addition, for this multi-class semantic segmentation task, the suggested model achieved state-of-the-art performance. Table 5 focuses on the DigestPath2019 validation set. The proposed model outperforms the FCN-8s, DeepLabv3+, PSPNet, SegNet, U-Net, and several versions of U-Net. Furthermore, the YAMU model achieves state-of-the-art performance for both the PANDA dataset and the DigestPath2019 dataset with the fewest number of learnable parameters. Future work will include a thorough examination of the proposed model on several types of digital image datasets for semantic segmentation in order to test the model's capacity across various distributions.

## 4. Conclusions

A novel architecture for more accurate semantic segmentation in histopathology images has been described. Our model's improved performance can be attributed to its revamped skip pathways, multi-scale input feature maps, and use of the deep supervision technique, all of which aim to improve context aggregation. We tested the proposed model on two datasets and found that it outperformed other state-of-the-art semantic segmentation approaches consistently.

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

## Appendix A.  Qualitative Results

We also assess the models on a qualitative level. The segmentation mask created by the proposed model and other state-of-the-art benchmark models are shown in Figure 3 & 4. Importantly, the proposed model's segmentation boundaries are finer and closer to the ground truth than DeepLabV3+ (with different backbones), which also includes multi-scale context information, as well as U-Net, U-Net++, and MSU, which use a substantially more complex model. This further reinforces our model's ability to extract meaningful morphological information through context aggregation.

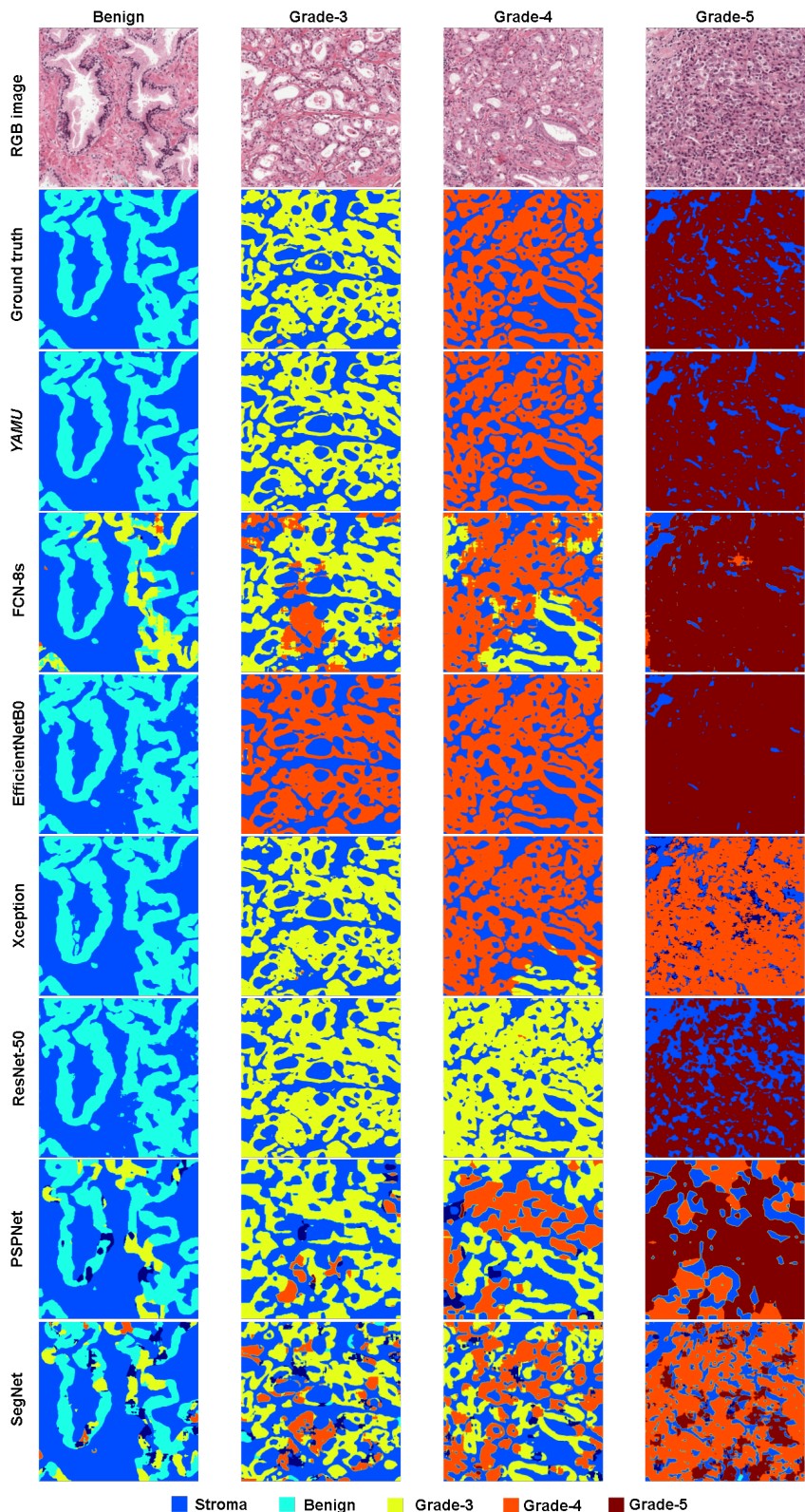

Figure 3: Comparison of tile label output for five different classes of state-of-the-art FCN-based models and proposed YAMU model.

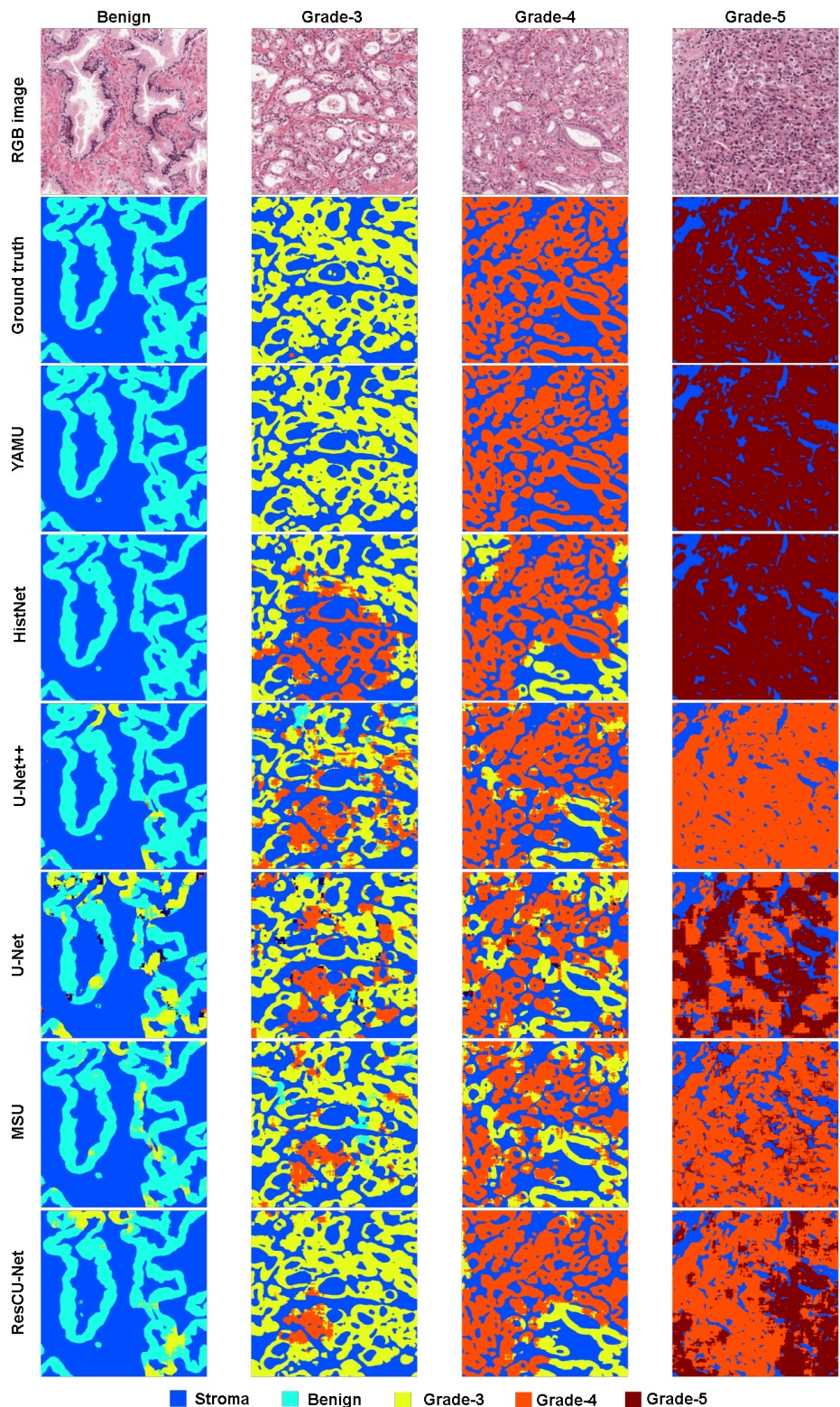

Figure 4: Comparison of tile label output for five different classes of state-of-the-art U-Net-based models and proposed YAMU model.

| Patch Size | Batch Size | Background | Benign | Malignant | Mean DICE |
|---|---|---|---|---|---|
| 256 | 32 | 98.82 | 74.87 | 86.04 | $86.57 \pm 9.78$ |
| 512 | 16 | 98.73 | 77.24 | 87.97 | $87.94 \pm 8.78$ |
| 1024 | 8 | 98.89 | 80.49 | 88.2 | $89.19 \pm 7.54$ |

Table 6: Proposed model performance with different patch size on PANDA2020 dataset

## Appendix B. Model Training

For the model comparison, two sets of training datasets were used, with the PANDA dataset having six classes and the DigestPath dataset having binary classes. The models were built using the Keras framework, which featured a Tensorflow backend. With Adam optimizer and step learning rate, we use an initial learning rate of $10^{-3}$ and a batch size of 8, decreasing the learning rate by a factor 0.1 every 150 epochs. All the models were trained using a sophisticated loss function, which is a weighted combination of categorical cross-entropy and dice loss function. The training continued for all the models until the loss converged.

Rotations in multiples of 90 degrees, as well as horizontal and vertical flipping, were used as augmentations. These were used before Hematoxylin-Eosin-DAB (HED)-light augmentations ((Tellez et al., 2019)), to ensure that the rotations and flip transformations of the images with and without HED augmentation were comparable. The purpose of HED-light augmentation is to transform RGB channel images to HED space using the colour deconvolution method in order to determine the contribution of applied stains based on stain-specific RGB absorption. For HED-light, we choose a 0.04 intensity value.

## Appendix C. Effect of varying patch sizes

We perform a study to determine the effect of varying patch sizes on model performance. Table 6 shows the results of this study. The patch size of $1024 \times 1024$ with batch size of 8 gave the best performance when trained on a NVIDIA RTX 6000 GPU.

## Appendix D. Stem block, Inception + module and Modified dense block

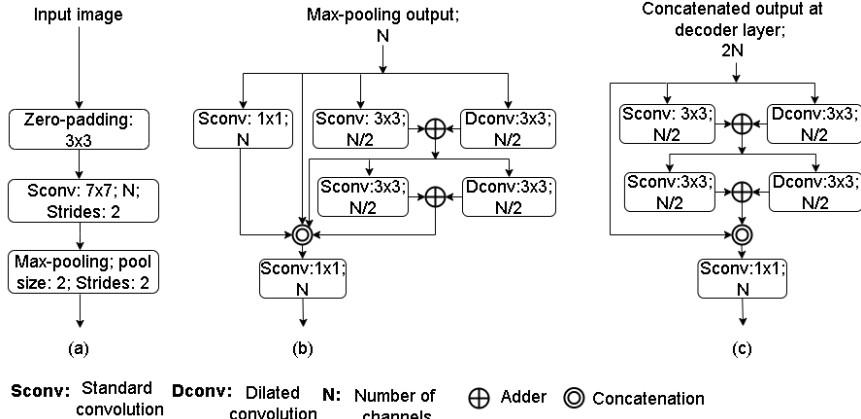

Figure 5: a) Stem block; b) Modified inception module using standard and dilated convolution namely Inception+ module used in YAMU-Net encoder layers $l_0, l_1, l_2$; c) Modified dense block used in the YAMU-Net decoder layer $D_1, D_2, D_3, D_4, D_5$.

