# OpenReview forum: "YAMU: Yet Another Modified U-Net Architecture for Semantic Segmentation"
_MIDL.io/2022/Conference — MIDL 2022_

### Official Review · Reviewer_ybEE · 2022-01-07

**Confidence:** 3
**Preliminary Rating:** 3
**Recommendation:** Poster

**Summary:**

The paper proposes a segmentation approach aimed at histopathology applications. It is an extension of the previous work of the author by multi-scaled feature maps (extracted from downscaled versions of the input image). Additionally, the model architecture was fine-tuned by using an auxiliary loss at lower resolution.


**Strengths:**

- The idea of incorporating multi-level features is interesting and already well established in histopathology.

- The paper combines some of the more recent ideas for segmentation models and evaluates them in a concise way.

- The model architectural decision all seem sound and the evaluation contains an ablation study.


**Weaknesses:**

The main weakness of this paper is that it was really hard to understand. This stems in part from the fact that it is strongly related to previous work of the same authors (HistNet), which represents an optimized version of a segmentation model aimed for histopathology.

To the reviewer it is not clear if the many modifications made against SOTA models are an overfit towards properties of the data sets or would be robust also for other data sets. Yet, the relatively small increments in performance given the large standard deviation in the ablation study could hint towards this.

As per table 1 of the original HistNet paper, it seems that the original architecture was tuned on the DigestPath2019 challenge dataset (test set). This somewhat restricts the possibilities to evaluate and compare against other methods where this was not done (could be perceived as a hyperparameter search on the test set). I have the same impression for this paper and the PANDA dataset (it’s unknown if the design choices were made on the validation set or on the test set, but the latter is my impression from reading the paper).

One of the data sets (the PANDA 2020 data set) is not yet released for own publications. The text on the website states:
"We ask challenge participants / researchers to respect an embargo until the challenge paper has been published. " (https://panda.grand-challenge.org/data/) Further, it is not clear if the evaluation was carried out on the test set of PANDA 2020 or the training set (and if, how this set was generated), or one of the sets available for evaluation on the kaggle platform.

**Deanonymize Review:**

no

**Detailed Comments:**

- Some parts of the introduction are verbatim copies or almost verbatim copies from the earlier paper of the authors. Under some regulations this is considered self-plagiarism. Please be cautious when copying from your old papers.

- I honestly don’t have a picture of your complete model in mind after reading the paper. In Fig. 1 the encoder block seem to be skipped. But how do you generate the downscaled multi scale input feature maps? I also don't understand how you can combine blocks of different feature resolution (yellow/blue/green/purple) without an upconvolution / convolution (which I don't find in Fig. 1).

- Please have in mind that the reader of your paper will typically not have read the HistNet paper beforehand. It would really help your impact if the paper would be understandable on its own without having to read your previous work.

**Paper Type:**

methodological development

**Questions To Address In The Rebuttal:**

- As some of the paper is hard to understand for this reviewer, it would be helpful to provide the source code of your model. Could you do that?

- You summarize that your model works better because of the architectural modifications you made. But since you apparently made the decisions knowing the results on the test set - how can you be sure it's not an architectural overfit to the data?

- Could you please rephrase the paper so that it can be understood without having read your previous ISBI paper?

- The PANDA data is not yet officially released (still under embargo). Isn't it a breach of license that you used it for your research?

- Did you only use the training set of PANDA? If yes, please indicate this properly in your paper.

**Special Issue:**

no

---

### Official Review · Reviewer_y2uB · 2022-01-23

**Confidence:** 4
**Preliminary Rating:** 2
**Recommendation:** Poster

**Summary:**

This work addresses the semantic segmentation task applied to digital histopathology imaging. The authors propose several architectural changes departing from their previous work and leading to improved performances and lighter memory footprint (i.e., less trainable parameters). The evaluation is performed on the PANDA and DigestPath2019 datasets.

**Strengths:**

The authors propose a modified u-net architecture allowing for higher performances and a lower number of parameters, which is appealing. The authors report a substantial amount of results, mainly on PANDA dataset, but also on DigestPath2019, which outlines the interest of the proposed method.


**Weaknesses:**

The proposed method is composed of multiple components and strategies, namely long skip pathways, multi-scale input feature map, and deep supervision training approach. This makes less obvious the understanding of the main contribution of the paper.

**Deanonymize Review:**

no

**Final Rating After The Rebuttal:**

4: Weak Accept

**Justification Of The Final Rating:**

I believe the authors did a good job addressing the comments and improving the overall quality of the paper. The appearing similarity of the metrics of many of the compare method prevents me from putting "Strong accept", as it is not obvious to judge the relevance of the proposed method. However, I believe the paper would raise the interest of the MIDL community regardless.

**Paper Type:**

methodological development

**Questions To Address In The Rebuttal:**

- In the related work section, the previous paper of the authors is discussed above all, leaving little room for other works. Some other works worth mentioning, in particular, graph-based methods https://arxiv.org/pdf/2107.00272.pdf. Could the authors extend the related work section?

- The presentation of the method is a bit difficult to apprehend due to the order of the sections. In particular, the datasets used appear to be introduced at the very early stage, before even the method is presented. Moreover, the architecture description seems to lack details. For example, the stem and inception+ blocks are not explicitly defined. Similarly, the architecture of the multi-scale feature maps does not stand out. Could the authors revise the Method section for clarity?

- Figure 1 contains multiple blocks of different colors. Unfortunately, very few details are given in the caption and the absence of the legend makes the reading even harder. Could the authors add a legend to facilitate the understanding?

- As part of the architectural modifications, the authors propose to modify the long skip connections. Could the authors discuss how it does relate or/and improve the performances or the behavior of the network compared to other proposed modifications of long skip connections (e.g., https://ieeexplore.ieee.org/document/8946755)

- In Figures 1 and 3 the Encoder layers are identified as E_i, however, later in Section 2.2.3, they are identified as l_i. Could the authors put some light on it?

- In section 2.2.2 three scales of the features are given, while only two kernel sizes. Is it an error, or is it intended?

- In section 2.2.3 on page 5, \sigma appears to stand for both ReLU and sigmoid operators. Could the authors rectify this?

- In the ablation study presented in Table 2, the deep supervision appears to be counter-productive when trained with P1 alone (row 5 vs. row 1). Could the authors detail that?

- The performances, as well as the number of parameters of the HistNet, reported in this paper differ from those reported in the original HistNet manuscript (https://openreview.net/pdf?id=zjiC7bqCAvv). Could the authors explain the differences if any?

- In section 2.1 The authors introduce two datasets being used. However, very few are being told about DigestPath2019 dataset. Could the authors put more light on the results from Table 5?

- The results in Table 4 seem to be different from the results in Appendix A (e.g., DeepLab with efficientNet). Could the authors provide more details?

Minor

- Could the authors revise the caption of Figure 2, specifically the phrase "It is recalibrate the encoder layer output".

- In section 2.1.1 Could the authors revise the sentence "The six labels in the annotation mask ...cancerous epithelium (Gleason 3), cancerous epithelium (Gleason 4), and cancerous epithelium (Gleason 5)" having three categories being cancerous epithelium?

- Could the authors proofread the text for minor typos?

**Special Issue:**

no

---

### Official Review · Reviewer_xwPP · 2022-01-24

**Confidence:** 5
**Preliminary Rating:** 4
**Recommendation:** Poster

**Summary:**

The paper describes a method for semantic segmentation in medical images. The proposed architecture is based on the U-Net and the previously proposed HistNet models. The authors modified the HistNet model by incorporating new skip pathways to the decoder part to aggregate multi-scale input features and a feature pyramid architecture. The proposed method was applied on two datasets and achieved superior performances compared to a number of deep learning-based approaches.

**Strengths:**

- The paper is well-written and easy to follow in most parts.
- The proposed method achieved a supervisor performance compared to the baseline HistNet and a number of deep learning-based models.
- The effectiveness of new proposed parts in the modified model is comprehensively investigated in an ablation study.

**Weaknesses:**

- The technical novelty and contribution in comparison to the previously proposed HistNet model is limited
- The effectiveness of the proposed deep supervision in the proposed workflow is questionable

**Deanonymize Review:**

no

**Detailed Comments:**

Please refer to the "Questions To Address In The Rebuttal"  and the "Weaknesses" sections.

**Final Rating After The Rebuttal:**

4: Weak Accept

**Justification Of The Final Rating:**

Thank you for addressing my comments and revising the manuscript. As the proposed approach achieved better semantic segmentation results, but its architecture is still similar to the already published HistNet model, I maintain a weak accept rating for this manuscript.

**Paper Type:**

both

**Questions To Address In The Rebuttal:**

- It is not clear why classes are reduced from 6 to 3 for the ablation study for the PANDA dataset.
- From the results in Table 2, it seems the proposed deep supervision method degraded the performance compared to the baseline model.
- It is recommended to publish the developed codes for the proposed model architecture.
- Please define HED in the appendix.

**Special Issue:**

no

---

### Meta-Review · Area_Chair_d5iC · 2022-02-20

**Recommendation:** Accept (Poster)
**Confidence:** 4

**Metareview:**

The paper presents an improved version of an existing model from the same group (HistNet), and shows some improvement across several datasets. Despite the fairly incremental contribution of this paper, reviewers generally pointed out that it has some value and that it might be interesting for the MIDL community.

---

### Decision · Program_Chairs · 2022-02-28

Accept